# “They Kept Going for Answers”: Knowledge, Capacity, and Environmental Health Literacy in Michigan’s PBB Contamination

**DOI:** 10.3390/ijerph192416686

**Published:** 2022-12-12

**Authors:** Erin Lebow-Skelley, Brittany B. Fremion, Martha Quinn, Melissa Makled, Norman B. Keon, Jane Jelenek, Jane-Ann Crowley, Melanie A. Pearson, Amy J. Schulz

**Affiliations:** 1HERCULES Exposome Research Center, Rollins School of Public Health, Emory University, 1518 Clifton Rd NE, Atlanta, GA 30322, USA; 2Department of History, World Languages and Cultures, Central Michigan University, 1200 S. Franklin St., Mt. Pleasant, MI 48859, USA; 3Department of Health Behavior and Health Education, School of Public Health, University of Michigan, 1415 Washington Heights, Ann Arbor, MI 48109, USA; 4Mid-Michigan District Health Department, 151 Commerce Dr, Ithaca, MI 48847, USA; 5PBB Leadership Team, Alma, MI 48801, USA; 6Pine River Superfund Citizen Task Force, P.O. Box 172, St. Louis, MI 48880, USA; 7PBB Citizens Advisory Board, Alma, MI 48801, USA

**Keywords:** contested knowledge, contested illness, PBB, oral histories, community mobilization, environmental health literacy, community capacity, community engagement

## Abstract

The Michigan PBB Oral History Project documented community residents’ descriptions of a large-scale chemical contamination—the PBB contamination—that occurred in Michigan in 1973. These oral histories document residents’ and others’ experiences during and after the contamination. We conducted a grounded theory qualitative analysis of 31 oral histories to examine the experiences of community members, researchers, lawyers, and others who actively sought out and contributed essential information about the contamination and its impacts. Our findings highlight several challenges encountered in the development of this knowledge including four central themes—contested knowledge, community skills, inaction, and uncertainty. Integrating environmental health literacy, community capacity, and contested illness frameworks, we examine the contributions of community residents, scientists (from inside and outside the community), and others to the development of knowledge to inform decisions and sustain action regarding this large-scale contamination. We close with a discussion of lessons learned regarding efforts to build environmental health knowledge within uncertain and often contested contexts and for promoting environmental health and action related to large-scale chemical contaminations. Our findings suggest the importance of integrated frameworks for examining and promoting the critical role of community skills, leadership, participation, sense of community, and community power in promoting environmental health.

## 1. Introduction

*I know science is slow to jump to conclusions about things, but … some of the things that I have seen and read that ended up being proven 20 years later, 30 years later, 40 years later. Because I appreciate that science wants to prove that theory. But a lot of times people are already there. They already know … those [cow] hooves were not normal … those hooves were deformed. And … they knew those animals, they were farmers. They knew what was right and what was wrong. And they kept going for answers, and people kept saying “no, no problems…”*.

-
*Michigan resident, PBB Oral History participant*


In 1973, the Michigan Chemical Corporation (owned by Velsicol) in St. Louis, Michigan, caused the largest food contamination event in U.S. history when it shipped flame retardants (polybrominated biphenyl, or PBB) instead of a nutritional supplement to be mixed into livestock feed. Farmers quickly noticed declines in their animals’ health, e.g., cows developed watery eyes, patchy skin, and poor appetites. Over time, more extreme effects were visible: joint issues, curved hooves, and spontaneous abortions. State agencies tested the feed and animals but did not know to test for PBB. Unable to determine the cause, state officials blamed farmers. However, a chemist-turned-farmer suspected contaminated feed and, nine months after the mix-up, confirmed the presence of PBB in the livestock feed by sending feed samples to laboratories across the country [1]. Shortly thereafter, the Michigan Department of Agriculture (MDA) began testing farm animals for PBB [2]. After initially claiming that only a few farms were contaminated, the state quarantined more than 500 Michigan farms and condemned approximately 30,000 cattle, 4500 swine, 1500 sheep, and 1.5 million chickens, as well as 800 tons of animal feed, 18,000 pounds of cheese, 2500 pounds of butter, 5 million eggs, and 34,000 pounds of dried milk products [3,4]. Contaminated farms that fell below the quarantine levels (1.0 ppm in May 1974, 0.3 ppm in November 1974, and 0.02 ppm in August 1977) faced tough decisions regarding the sale of their goods into the marketplace. Researchers estimated that eight million Michigan residents consumed PBB-contaminated farm products [5]. At the time, while there was little formal science about the health impacts of those exposures and when uncertainty bred inaction, inaction bred distrust. Farmers and Michigan residents faced the loss of animals and livelihoods, experienced emotional, economic, and social turmoil, and in the time since, a myriad of health concerns [2,6].

Decades later, researchers have confirmed what farmers observed, demonstrating heightened health risks associated with PBB exposure, including thyroid problems [7,8], lower Apgar scores at birth [9], earlier menarche [10], increased risk of miscarriages and breast cancer in women [11,12], and urogenital problems among men and boys [13]. In the immediate aftermath of the PBB disaster and in the decades that it took to build this body of scientific evidence, residents of affected Michigan communities struggled to comprehend, articulate, and take action in response to their PBB exposures and health impacts.

Michigan residents seeking to document and understand this nearly 50-year toxic legacy partnered with researchers in 2018 to conduct oral histories. The oral histories document the wide range of actions in which residents engaged as they sought to understand and make sense of the PBB contamination. The experiences of those who have lived through large-scale chemical contaminations are critical to inform both prevention of, and response to, future contamination events.

These experiences also offer important lessons to inform our understanding of environmental health literacy (EHL). EHL is an emerging area of study that incorporates content and strategies from environmental, health, and social sciences to promote understanding of the ways that people build knowledge about environmental contaminants and take action to protect health [14,15].

While many definitions of EHL tend to emphasize the role of environmental health scientists and educators in empowering individuals and communities [15], the PBB contamination in Michigan offers a unique opportunity to examine the role of the exposed individuals and communities themselves in the process of knowledge building and self-empowerment. Our analysis examines the efforts of exposed individuals and communities to understand the contamination, develop knowledge, and take action to protect and promote health.

In this paper, we present the themes that emerged from our analysis of community members’ experience with the PBB contamination. We examine these themes within the context of relevant theoretical frameworks to inform our understanding of environmental health literacy, community capacity [16], and contested knowledge [17]. Understanding the processes through which Michigan community residents acted to protect their health and well-being offers much to the field of environmental health. Our hope is that the experience of Michigan residents will inform both the field of environmental health literacy, and the efforts of scientists and others to effectively translate environmental science into action.

## 2. Materials and Methods

We utilized a qualitative case study approach [18], which included collecting oral history interviews with individuals who had first-hand experience with the PBB contamination in Michigan during the 1970s. This approach is useful in gaining an in-depth understanding of an event or complex phenomenon. Interviews were conducted as part of the Michigan PBB Oral History Project (https://scholars.cmich.edu/en/publications/the-michigan-pbb-oral-history-project-7, accessed on 11 March 2022). The purpose of this ongoing project is to gather and preserve the stories of individuals affected by the Michigan PBB contamination and create a special collection or repository of their stories. Oral history methodology [19] was used as it provides a systematic process for identifying, exploring, documenting, and analyzing complex community experiences and/or events. Appropriate Institutional Review Board approval or exemption status was obtained from each of the three universities participating in this study. Notably, this project was initiated by the PBB Leadership Team, consisting of two representatives from each of the PBB partners: the PBB Citizens Advisory Board, the Pine River Superfund Citizens Task Force, the Mid-Michigan District Health Department, and Emory University PBB Research Team. During monthly meetings, the PBB Leadership Team provided input into the PBB Oral History Project from proposal development to project management.

### 2.1. Participants and Recruitment

Project staff began by creating a list of more than 100 individuals with important insights and experiences with the PBB contamination. These individuals were identified through various means including prior participation in PBB research, attendance at community meetings for the Michigan PBB Registry, and referrals by members of the PBB Citizens Advisory Board and Pine River Superfund Citizens Task Force. Also included on this list were individuals who contacted the project team after seeing media coverage of local community meetings. Project staff contacted these individuals via email and telephone calls to obtain additional information on their experiences, screen them for potential inclusion in the oral history project, and invite them to be interviewed. A total of 68 interviews were conducted. The analysis presented here is based on the first 31 interviews completed. This subset was chosen because at the time of the analysis, these interviews: (a) had been reviewed by participants and approved for donation to the Michigan PBB Oral History Project’s special research collection/repository; and (b) represented a mix of individuals from varied perspectives and experiences with the contamination (e.g., farm families, residents and other consumers of farm products, chemical company workers, attorneys, veterinarians). 

### 2.2. Data Collection

Interviews were conducted between August 2018 and June 2019. Three interviewers who were trained in oral history interviewing methods conducted all of the interviews, including one senior researcher (BF) with several years of experience in oral history methods. In most cases, interviews were conducted in person by an interviewing team of one senior interviewer and one junior practitioner. In addition, interviewers communicated with each participant prior to the interview to collect important background information that helped them (a) identify the appropriate set of open-ended, semi-structured interview questions, (b) expand the chosen set of questions to gain additional insight, (c) establish rapport with participants, and (d) explain the interview processes to participants, including informed consent and future use of data. Semi-structured interview guides provided flexibility in how interviewers approached key subjects, as well as provided opportunities for follow-up questions that led to the discovery of new and previously undocumented information. Interviewers used open-ended, stakeholder-specific questions that reflected the variety of PBB community experiences, as well as common questions to establish a baseline in the dataset. For instance, interviewers asked each participant: “what do you remember about the PBB mix-up in the 1970s?”, “what did you learn about possible health effects?”, and “what do you hope others might learn from your story?” Most participants were interviewed in their homes, although a few were conducted in quiet public places (i.e., libraries, campus offices). Interviews with two participants took place on two separate dates, and when this occurred, transcripts from both dates were combined into one for purposes of the analysis. Oral histories varied considerably in duration, with the average length of an interview lasting 1:21:34 (range: 26:50–3:36:40). These methods sought to create an interview environment in which participants felt comfortable sharing personal and often painful memories. All interviews were audio recorded, transcribed, de-identified, and verified for accuracy by the project team.

### 2.3. Data Analysis

We performed content analysis on interview transcripts using a grounded theory approach [20]. Team members began by reading and re-reading the first five interview transcripts independently to become familiar with the data, note initial ideas and sensitizing concepts, identify domains of interest, and ensure content immersion. The team then started constructing a preliminary codebook using the study aims along with the identified domains of interest and sensitizing concepts. Five team members with experience in qualitative analysis then selected the same transcript to independently review, and applied both deductive codes and inductive codes to the text. Team members then met via Zoom video conferencing to discuss the coded transcript, reviewing line-by-line the text and corresponding codes. Differences in applied coding were discussed and agreement was reached by consensus. The codebook was revised and then applied to subsequent transcripts. The next 20 transcripts were then divided and assigned to various members of the team, with two researchers independently coding each transcript, and meeting to discuss any discrepancies until agreement was reached. When new codes were developed, the codebook was refined and previously reviewed transcripts were re-reviewed to apply the new codes. This process increased inter-coder reliability and ensured consistent coding across the dataset. After coding was completed for 21 transcripts, a check of inter-rater reliability was conducted and a rating of “very good” was achieved (kappa coefficients k = 0.75–1.0). The remaining 10 transcripts were coded by one of the team members (rather than two), following protocols suggested by Fleiss, Levin, Paik [21]. NVivo 11 software (QSR International Pty Ltd., 2018, Burlington, MA, USA) was used to assist with data management and to develop code summary reports. These code reports, which included all data aggregated under the same code, were generated and analyzed by the larger study team to identify and discuss patterns in the data and overarching themes. Team members met after reviewing each code report. Rigor was increased through the involvement of multiple team members in all phases of analysis, interpretation of data, and development of themes.

The themes presented here represent a subset of overall themes that emerged in this analysis, focused on our central research question of environmental health knowledge to support individual and collective action. These themes were identified and named in a two-step process. First, we created a short descriptive name, remaining as close to the language used by participants as possible, and then selecting a brief quote that captured the concept included in that theme. Next, these themes—described here as subthemes—were clustered into larger themes, following processes described in the grounded theory literature [20] to create broader themes. These broader themes were then named using the same two-step process. We selected direct quotes from the oral histories to illustrate each theme.

This paper’s results are organized using the themes and subthemes that emerged from this process. In our discussion section we examine the themes that emerged from our analytic process in the context of three conceptual frameworks related to environmental health and action: environmental health literacy, community capacity, and contested illness. Finally, in an effort to integrate our findings with the existing literature, we propose a conceptual framework that builds on our analytic themes and the existing conceptual literature.

## 3. Results

We describe central themes, subthemes, and cross-cutting themes that emerged from analysis of the oral histories, with illustrative direct quotes from the oral histories. Central themes and subthemes reflect the contestation of community knowledge; efforts to seek information, and build knowledge and understanding; human impacts of uncertainty; implications of inaction by presumably trustworthy institutions; and emergence and maintenance of community action. Cross-cutting themes interwoven through those central themes include *uncertainty,* associated with lack of information or contested interpretations; *(in)action*, defined as actions taken (or not taken) in the face of the environmental contamination; and the mobilization of *community skills and resources*, defined as human or organizational capital to address the environmental health issue. These themes are outlined in Table 1 and described in the sections that follow, including illustrative quotes directly from participant oral histories. Table 2, Table 3, Table 4 and Table 5 include additional illustrative quotes for each subtheme. All de-identified quotes are presented verbatim to reflect the participants’ voice, including necessary clarifications in brackets. We retained the names of prominent actors whose names are part of the public records associated with the contamination.

### 3.1. Theme One—Contesting Community Knowledge: “They Were Telling People We Didn’t Know What We Were Talking About.”

This theme captures the experience of many farmers and other community members in the early days of the PBB contamination, when farmers’ own observations and descriptions of changes in their animals were challenged and disputed by industry representatives and trusted institutions. In some cases, farmers were blamed and told that the problems were due to their own farming practices or the way they cared for their animals. As these farmers continued to seek information and answers, they faced several challenges. Those included multiple delays in action by government agencies across multiple levels, contributing to farmers’ frustration and anger. The crosscutting themes of *community skills and resources, uncertainty*, and *(in)action* appear throughout theme one: Farmers’ skills are apparent in their observations, while their knowledge was contested and undermined, contributing to uncertainty and inaction by both individual farmers and decision makers (See Table 2).

#### 3.1.1. Personal Observation of Animals: “There Is Something Wrong.”

Michigan farmers who participated in the oral histories described early signs of the PBB contamination in Michigan, observations informed by their deep expertise in farming, animal care, and milk production. Those observations informed their hypotheses regarding the source of the problem, with, for example, one farmer describing their early suspicions: *“I think there’s something wrong with what I’m giving them as nutrient.”* Another farmer described the profound emotional impact of these early observations: *“… it seemed like it came on quite fast … our second son … I think it hit him the hardest of any of our children, and today his heart aches for those cattle … He said ‘Dad, I can’t—it just breaks my heart to pick those calves up and massage their legs, and drench them and just have them go back down next morning, and they’re gone.’”*.

#### 3.1.2. Contested Nature of Knowledge about the Contamination: “But Yet We Couldn’t Make Anybody Understand.”

As farmers attempted to understand what was causing the morbidity and mortality amongst their animals, they sought answers from trusted institutions. For example, one farmer who had learned of the distribution of contaminated animal feed, described their brother asking Farm Bureau Services, *“‘could we have possibly got any of this pelleted feed that they are concerned about?’ They assured us that no, there was no way we would have got it … So, we took them at their word and went on with our business.”* Instead, as one participant described, *“… the Farm Bureau kept coming back saying, ‘Well it’s the way you’re treating your cattle. It’s your animal husbandry.”* Another participant made a similar observation, noting, *“But I remember just being in shock at, at how the farmers had been lied to or blamed and that nobody wanted to admit that—or look into, why is this happening in so many places?”*.

These actions, which contested farmers’ own observations and knowledge, also undermined the development of a shared understanding of the contamination and its impacts. Ultimately, the resulting uncertainties contributed to delays in action. Farmers’ dismay at being blamed for the deterioration of their animals’ health soon gave way to frustration and anger.

#### 3.1.3. Inaction and Frustration: “They’re Not Doing Anything!”

As the Farm Bureau Services and other institutions (e.g., MDA) failed to recognize or act on the widespread contamination, faced with the loss of their livelihoods, farmers increasingly gave voice to their frustration. As one resident close to the situation shared, *“here is Farm Bureau waffling, and here is the state legislator, they attended meeting after meeting after meeting after meeting. At a couple of which farmers got up and shouted ‘They’re not doing anything!’ you know. The farmers absolutely, they were appalled, dismayed, discouraged by nobody coming to them.”* The dismay and anger that emerged out of these initial efforts to understand the emergent environmental health crisis, and the delays in action, contributed to tension and conflict, examined later in this section. Eventually, this frustration and inaction led community members to mobilize their own resources to better understand the scope and implications of the contamination.

**Table 2 ijerph-19-16686-t002:** Central Theme One—Contesting Community Knowledge: “They were telling people we didn’t know what we were talking about.”.

Subthemes	Additional Illustrative Quotes	Relevant Cross Cutting Theme(s) ^a^
Personal observation of animals: “There is Something Wrong.”	At first, I remember thinking—people would say, “There must be something bad in the feed.” And I’d think, “That wouldn’t happen. That would not happen! That they’re poisoning—you know, that poison got in there.” But, that’s exactly what happened. And it just made lots of hardships because, with most of the people being farmers, their income was reduced. They had trouble paying their insurance. They had trouble buying groceries. You know? It trickled down to everybody. (Resident) I knew I had something that I’d never seen before. I knew I had something that was strange. It—that’s the best word for it—it was strange. A lot of the symptoms that my colleagues were mentioning I did not see. The enlarged feet, the skin problems, we did not see that in [redacted] herd. Our biggest problem was abortion. The cattle were losing their calves. What calves were born were deformed, they had frozen joints—they were very difficult to maneuver inside the cow and to even deliver them—they were so deformed. I knew we had something that was completely different. (Veterinarian) Then [1974] came along, and all of a sudden, our cattle weren’t doing worth a hoot. As an example, before PBB we had an 18,000 pound plus herd average, pound per pound, per cow per year, which was in the top ten in the county, so that was decent. And within six, seven months’ time it dropped down to just over 9000. We were treating them the same way, feeding them the same way, doing everything like we always did, nobody could figure out what was the matter with them. They, they—they just—some of them died, but a lot of them—most of them didn’t. The young stock—the calves wouldn’t grow, the cows wouldn’t rebreed and reproduce—I mean they just didn’t, no matter what we did and we never had that problem before; so, we were all frustrated… (Farmer) He knew something was wrong. Where we kind of hoped it wasn’t something we were doing, but we couldn’t figure out what we were doing wrong. And well then when it came up you know, hey there’s a problem here. That was probably the biggest relief, (pause) that we could ever hear, was that (sighs), it’s not, you know, it’s not your fault. (Farm family)	Community skills and resources
Contested nature of knowledge about the contamination: “But yet we couldn’t make anybody understand.”	Cattle had different symptoms. It was so elusive. You couldn’t put your finger on it. And it was always—it started out, “That material went to these farms, and this particular feed in Farm Bureau had it in.” Not realizing it went through the mill, and all these other feeds got contaminated. And so originally we said we didn’t get it. We didn’t feed that Farm Bureau fifty-five percent protein. We didn’t feed that. So we said we didn’t get it. And when they found out that we did have it—the defense made the case that it was poor nutrition. That the farms that were complaining had poor nutrition, and they were poorly managed. (Farmer) It’s just like what Farm Bureau did with PBB originally. “I’m sorry guys,” you know, “it’s not PBB, there’s no such thing as PBB contamination. This is not—this is bad farming or this is whatever,” the government denied that there was a problem with Farm Bureau backed them up for a long time, you know. If you don’t say anything about it, it won’t—it will go away. (Resident) So there really wasn’t anybody. And then we had Michigan Chemical. Farm Bureau. The MDA. The Health Department. The only person, … is [name redacted] who believed in us. And, tried to help us. But then we had a senator … cause we tried to tell him that it’s not just the farmers, it’s going all over Michigan. Canada closed off our cattle from going over there. They were smart. Cause they believed it. But yep, we couldn’t make anybody understand that it was not just the farmers. (Farm family) And then you saw the state kind of backing off from listening to what the citizens were sharing with them. I don’t know if they listened. Did anyone listen to the people, workers at the chemical plant, or farmers? I can’t answer that, but I have the feeling that no one really listened to them. (Resident) Anyway, we found, we started having trouble in the cows, and that was a pistol because we were shutdown at every pass. Because of this, actually Farm Bureau didn’t want us to know what happened. Michigan Chemical didn’t—Northwest Industries didn’t, you know, they didn’t—and they called me kind of some goofy names. They weren’t nice. (Farmer)	Uncertainty
Inaction and frustration: “They’re not doing anything!”	There was a lot of anger in the community. ‘Why did this happen? How could this have happened? And why did it take so long to be discovered when there were so many sick animals and dying?’ (Resident) So, there is where you got your widespread contamination in the people. It didn’t get stopped. Nobody ever put a stop to it. And that is why you are having your study today, because it got spread to everybody in the state, and it even went out beyond. (Farmer)	(In)Action

^a^ Cross-Cutting Themes: *Uncertainty*, associated with lack of information or contested interpretations. The mobilization of *community skills and resources*, defined as human or organizational capital to address the environmental health issue. *(In)Action* defined as actions taken (or not taken) in the face of the environmental contamination.

### 3.2. Theme Two—Seeking Information, Building Knowledge and Understanding in the Midst of Uncertainty: “So Anyway, It Was a Tough Deal. It Was Tough Trying to Figure It Out.”

Oral history participants described multi-pronged efforts to gather information and develop an understanding of the emergent PBB crisis. Community members leveraged their own experience, skills and expertise to understand the contamination and its health impacts. They also sought to build systematic information in conjunction with medical and environmental health researchers outside of the immediate community. Yet, the scientific information had its own limitations and uncertainties. Here, the crosscutting themes are represented by community members utilizing their own scientific, veterinary, and information seeking *skills and resources* to build knowledge, yet being faced with *uncertainty* and *inaction*. While presented sequentially below, these themes overlap chronologically, reflecting knowledge building as a reflexive and iterative process (See Table 3 for additional illustrative quotes).

#### 3.2.1. Leveraging Community Expertise to Build Knowledge: “Do Your Own Research and Don’t Give Up.”

Frustration with the inaction, combined with many farmers’ disbelief at being blamed for the symptoms that they were observing in their animals, led many to redouble their efforts to build their own understanding. For example, one participant described *“… we had the powerhouse [Farm Bureau Services] telling us we didn’t … know what we were talking about. So, we weren’t very thrilled about it. But you still had to go with what you believed, you know?”* Another described how farmers reacted to this response by seeking their own information: *“Go with your gut instinct, get the hard facts, the true science about it. Do your own research and don’t give up … If you think you’re right and you’ve got that gut feeling, go until they prove you’re wrong.*” Notable among early efforts to seek information were those of Rick Halbert, a local farmer and chemist who noted adverse effects among his cattle. One interviewee noted that *“Rick said ‘There is something wrong here’ … he sent [samples of cattle feed supplement] off to an independent lab and it came back and there was [PBB] in there.”* (The referenced lab was the USDA Agricultural Research Lab in Beltsville, Maryland. They discovered the presence of Firemaster bp-6 in the feed sample because one of the scientists left a mass spectrometer running hours longer than normal and it indicated on the readout a “brominated ring compound.” (per a personal note that Dr. Alpha “Doc” Clark’s wife wrote documenting a phone call from Rick Halbert to Dr. Clark on 4-22-74), available from the Central Michigan University Clarke Historical Library: https://quod.lib.umich.edu/c/clarke/ehll--clarkalpha?cginame=findaid-idx;id=navbarbrowselink;subview=standard;view=reslist, accessed on 11 November 2022)).

Others also used their skills to understand the extent of the contamination in its early months. For example, this interviewee described the role of local Veterinarian Dr. Alpha “Doc” Clark in gathering fat samples from dead cows as evidence to help establish the extent of the PBB contamination and links to livestock deaths: *“… He said, ‘Just did another cow … this cow just dropped over today. ’And so he took the fat sample right there.”* A former state employee, after being contacted by a local farmer regarding sick chickens, described checking to see if there was PBB contamination: *“And I looked at the chickens, and they were miserable looking animals. They just looked awful. And I thought to myself, ‘… I wonder if there’s any chance they’ve been fed the contaminated feed around—from around the state…’ And sure enough they examined the eggs and they had low levels, but certainly measurable levels of PBB in them.”* This individual then independently reached out through his networks to explore further the extent of the contamination in humans: *“So I asked the pathologist if they would obtain a couple of fat samples from various people for me. And I sent those to the laboratory, and all four of them had PBB. So I knew it was a widespread problem throughout the state.*” These efforts were instrumental in moving toward more formal and geographically dispersed state testing for PBB, and in drawing attention to the need for systematic decision making.

Community members also drew upon their information-gathering skills to document the contamination. One participant described gathering documentation related to every aspect of the unfolding PBB crisis: *“So anyway, it was a tough deal. It was very tough trying to figure it out. But in the end … I ended up with twelve boxes of material.”* This community scientist documented events that unfolded over decades. Public hearings, decisions, policy statements, and community meetings contributed to understanding the source of the contamination, its impacts, and informing the actions of residents and decision makers.

As evidence emerged regarding the widespread nature of the contamination, concerns about the health impacts of consumption of potentially contaminated meat and dairy products grew. Many turned to researchers from outside the community to help in this process.

#### 3.2.2. Seeking Information Outside the Community: “So I Am in That Study.”

Residents’ descriptions of their engagement with outside researchers to help understand the contamination included their participation in studies, as well as the resulting challenges in obtaining, and then interpreting, that information, described in the following sub-themes (underlined).

##### Seeking Outside Expertise, Testing and Studies: “So I Had My Blood Taken.”

Even as community members built their own body of evidence, they also sought opportunities to participate in studies conducted by outside researchers, sometimes over decades: *“I am in the Michigan Department of Health lifetime [study], so they’re monitoring all my health records and stuff, for the rest of my life. And I don’t know, if I die … I would donate my body to have it biopsied, or there whatever they would want to do to see what PBB did to me.”* (Note that the Michigan health department is no longer continuing the monitoring, however Emory University continues the health research).

##### Inaction and Challenges Obtaining Information: “It’s Always Surprising to Me to See How People Were Unable to Get Information.”

Participants described multiple challenges as they sought out opportunities to obtain testing. One participant, convinced that her daughter’s newborn child was affected by in-utero exposure to PBB, described her experience with the health department: “… *They took her in one room … and said, ‘Do you think this is PBB?’ and … she said, ‘I have no idea.’ … And they said, ‘Well it costs a lot to get tested and we’re not gonna to do it.’ When I told my husband he said, ‘She will be tested if it takes the last penny we’ve got. She will be tested.’”* Another participant reflected on this situation: *“It was always surprising to me to see how people were unable to get information. People who wanted to be tested couldn’t get tested. There wasn’t a government agency coming in to do the work.*” Several participants who were able to enroll in research studies described their disappointment in not receiving study results: *“Because in the past there have been people that have come and done studies, and then we don’t hear back from them. So, we don’t know what the results were.”* Difficulties obtaining information or findings from the studies conducted were compounded by challenges interpreting the results.

##### Scientific Uncertainty: “Well We Really Don’t Know What It Means.”

Individuals who sought answers by participating in various tests and studies were hampered by the fragmentation, delays, and uncertainty of the science. One participant described inconsistencies in results from different studies over time, *“… I got back blood tests in 2012 that said that there was no detectable polychlorinated biphenyls there, that’s PCB, and polybrominated biphenyls [PBB], … Then I hear in 2017 about the PBB Registry project and I submit my blood and then just this year I get back the test results showing, ‘Yes indeed, you do!’”* This same participant also described uncertainty in interpreting their initial results, referencing the state’s quarantine level at which a cow would have been euthanized and buried in a state-operated landfill: *“I’m just looking now at my first report that says I have 1.16 part per million in my fat, … So, if I was a dairy cow I’d be up in the landfill, they’d shoot me and bury me*.” Over 30,000 cattle were destroyed based on their PBB levels [3,4,21], while uncertainties remained regarding the levels found in humans and what actions could be taken based on that information. The slow pace of science also contributed to ongoing uncertainties: *“I asked them ‘well what does all this mean?’ And he says ‘Well we don’t really know what it means. You’ll have to come back 40 years later and we probably will have some answers for you.’ Well, here it is 40 years later and yeah, it does do some terrible things for yourself.”*.

The slow pace of the scientific process and scientific uncertainties were in tension with the urgency with which community members wanted and needed answers, regulatory action, and health advisories to prevent further exposures and adverse health impacts. For example, this resident and former university researcher expressed frustration with the delays that could have prevented exposure via breast milk: *“… in 1974 I had raised concerns about women breastfeeding their babies. By June of 1976, the Michigan Department of Public Health announced that—Oh they found PBB in human breast milk! (both laugh) So two years—almost about two years later*.”.

Inconclusive evidence and contested explanations for the contamination and human health implications contributed to delays in action on the part of the state’s regulatory and health advisory organizations, and continued uncertainty among residents.

**Table 3 ijerph-19-16686-t003:** Central Theme Two—Seeking information and building knowledge and understanding in the midst of uncertainty: “So anyway, it was a tough deal. It was tough trying to figure it out.”.

Subthemes	Additional Illustrative Quotes	Relevant Cross Cutting Theme(s) ^a^
Leveraging community expertise to build knowledge: “Do your own research and don’t give up.”	I thought the only way there’s going to be a resolution is if St. Louis, Michigan [site of Velsicol plant] gets a health study so someone can say definitively, “Yes, there is something wrong here and they’re not living the same quality of life and maybe they’re not living as long … So, I started a health questionnaire. (Resident) Cause I remember for a long time we had a cow’s leg in the freezer. We were keeping that from one of the contaminant. We were keeping that for—by god that was data down there. (Farm family) What the state was doing when they found PBB, they put it under the 0.3 so they didn’t have to pay no money. The companies did. And I thought, “Well hell, this don’t sound right.” So I start sending split samples to WARF. That’s how it went. And I met the doc—a guy over there and he was a super guy. And he—he told me he knew what a mess I would be in to. But he, I think we sent, and I might be wrong, but I think we sent about twenty-five samples over there … We did the right thing. Not that we’re so smart, but we did the right thing. (Veterinarian and Farmer) I called the Michigan Chemical Company and asked them for a sample, so I could do some studies. And asked them if they had done any testing, they said, “No. It’s perfectly safe as far as we’re concerned.” And so they sent me a sample of the Firemaster BP-6. And—um—so I fed it to some pregnant mice. And I thought, “Somebody’s got to do something.” Cause there were no—absolutely no studies in the literature (pounds table) about any toxicity studies (pounds table) that had been done on this chemical. And I thought, “This has horrendous potential.” (Doctor)	Community skills and resources
Seeking information outside the community: “So I am in that study.”	And I don’t know if I found out about it through the newspaper or how I found out about it, but there was this study through Michigan State [University] where they were testing mother’s breast milk to see if we had PBB in our breast milk. And so I remember—so I sent several samples down to Michigan State to be tested. (Resident) … when they initially did all the testing in Lansing [Michigan], my oldest sister refused to do it, cause she didn’t want to know. The rest of us went through it, and my brother and I ended up, if I remember correctly, being at the highest levels in our family. Because we drank the most milk. So. Yeah. (Farm family) But the reason I know what the exact results were is that I entered what they were in my son’s baby book. (both laugh) So I have a copy of my baby book page here, somewhere yeah. Okay. Where I entered, at the time, exactly what the PBB tests results were. (both laugh) And we got those when he was five and a half months old. (Resident)	Community skills and resources (In)Action Uncertainty

^a^ Cross-Cutting Themes: *Uncertainty,* associated with lack of information or contested interpretations. The mobilization of *community skills and resources*, defined as human or organizational capital to address the environmental health issue. *(In)Action,* defined as actions taken (or not taken) in the face of the environmental contamination.

### 3.3. Theme Three—Human Impacts of Uncertainty: “They Have Had Issues and Doctors Can’t Explain It.”

Participants described the profound human impacts of living with uncertainty. These included the challenges associated with making health-protective decisions for themselves and their loved ones in the absence of information, and the emotional and physical toll caused by the knowledge that decisions made or not made in the face of those uncertainties had profound implications for health and well-being. See Table 4 for additional quotes for each subtheme.

#### 3.3.1. Uncertainty and Ambiguity Regarding Health Guidance: “They Basically Could Not Get Any Solid Information from Anybody.”

Scientific uncertainties had implications for personal decision-making. Participants described the inability of medical professionals to make diagnoses: *“They have had issues and doctors can’t explain it. They tried them on this, or they tried them on that or … ‘well this doesn’t show, so you don’t have this, so you must be OK’.”* The lack of information hampered their ability to make health-protecting decisions. For example, one participant described her daughter’s struggle to decide whether to continue to breastfeed her infant, *“Should she keep nursing? Should she stop nursing? And they basically could not get any solid information from anybody … They said … ‘we don’t have any results. We don’t have any conclusions. We’re just in the beginning stages here.’”* Clear and accurate information was necessary, and the lack thereof was a barrier to action: *“… at that time there was just limited things that you could do … the biggest thing is information. Accurate information, correct information, no guessing, no surmising, no what-you-think, or opinions, or whatever, you just needed to keep the facts straight and stick with the facts.”*.

Beyond limitations on the ability to make informed, health protective decisions, the physical, emotional, and social toll exacted by those uncertainties are examined in the following subtheme.

#### 3.3.2. Physical and Emotional Impacts of Uncertainty: “I Think of All Those Missteps, Just from Lack of Knowledge.”

At the time that these oral histories were conducted, over 40 years after the PBB contamination occurred, participants continued to describe the emotional impacts of the contamination, their contested knowledge, and the associated uncertainties. One member of a farm family described the emotional and financial toll of not being able to sell their agricultural product while being responsible for the health of others: *“… it was devastating you’re doing all this work and you’re dumping the milk and you can’t sell the animals. And you’re like, stuck in this limbo … I don’t want to poison anybody, and it’s like we were placed in a situation, not by any of our own choice.”*.

Another participant described consuming foods during her pregnancy in the belief that they would help assure a healthy child, yet learning later that many of the products she consumed had been contaminated with PBB: *“I ate more meat then than I ever did. I’ve never been a milk drinker, but while I was pregnant I drank milk because I wanted to build this baby, this strong baby. And I think of all those missteps, just from lack of knowledge, lack of information, and lack of transparency by the government here.”* As one participant noted, *“We were all fearful. (There was) a lot of uncertainty at that time* … *It basically affected everybody, even if you weren’t a farmer, because you worried about what you were eating and drinking.”*.

Many Farmers also described the emotional impacts of having their own observations and lived experience discounted: *“It was horrible because … you knew that most of what you knew was accurate and they were telling you that you didn’t know what you’re talking about … and pretty powerful people telling me that.”* One participant who was a child at the time reflected on the stress and tension in conjunction with the economic impact his family experienced: *“I remember losing cattle, I remember them being sick, I remember them coming with a pickup—you know, trailers, and hauling them off … it was traumatic … I’ve heard my mom talk a ton about when the cows started getting sick, how she lived with migraines on a daily basis because she was so stressed out … Until, you know, we lost the farm … And that was their entire livelihood.”* The tension and stress that reverberated within families and communities had profound emotional impacts: *“No one talks about the collateral damage of the event. We always talk about how many thousand head of cattle and sheep and eggs and so on and so forth, but the collateral damage is the farm families themselves. The parents that got in arguments and got divorced, the kids fighting with their dads that left the farm. A lot of anxiety, depression, worrying—unnecessary worrying—these are all the collateral damages that nobody speaks about, and they’re just as real as the monetary things.”*.

Uncertainties and the high economic stakes for farm families also contributed to damage to longstanding social ties. A participant who received a settlement for having his contaminated cows destroyed, described the reaction of some neighboring farmers: *“… there was a few loyal Farm Bureau members in the area that were also farmers … one of them came to see me and accused me of just looking to make a killing … That was tough, these were people I grew up with and knew my whole life…”* Ambiguous information and contested knowledge affected the health and relationships of individuals faced with making decisions to protect their health, their families and their livelihoods.

The PBB contamination both mobilized community action, as described in previous sections, and undermined community characteristics central to the ability to take action, as captured in the above quote. In the following section, we examine their implications for trust in broader social and political institutions.

**Table 4 ijerph-19-16686-t004:** Central theme three—Human impacts of uncertainty: “They have had issues and doctors can’t explain it.”.

Subthemes	Additional Illustrative Quotes	Relevant Cross Cutting Theme(s) ^a^
Uncertainty and Ambiguity regarding health guidance: “They basically could not get any solid information from anybody.”	And I know when my first child was born—this is with the State of Michigan—I had to send in four specimens, my own blood, a fat biopsy, cord blood, and breast milk. And so we sent those specimens off, but what we never heard back from them about was that any comparison to other people that were being tested. Or where were we on the curve of lower contamination to higher contamination. (Resident) Hard to ascribe cause and effect but there was a significant impact on morbidity and mortality in the community and the community didn’t—wasn’t able to get answers about. (State researcher) Dr. [redacted] next door is the one who really just sprang into action. And he was—I would talk to him practically every other day where he would tell me who he contacted and how he couldn’t get any answers from anybody … Well back in 19—I don’t know what tests were, what things were done between 1973 and 1977. But when U of M did that breastmilk study in 1977 I was living next door to a really smart man, Dr. [redacted]. And he was just bound and determined to get information on this. And he made phone calls and inquiries all over the place. He contacted the Center for Disease Control, he was contacting everybody in Michigan and when he couldn’t get answers from anybody he went to the federal government level. He tried mightily to get information … And, you know, his notion was you develop problems, you go to work on these things and start gathering the information. And he was appalled that he could not get—that a medical professional trying to treat patients, okay, who were known to be—have the stuff in their bodies—could not get information that would be helpful to him. (Resident) So I’m thinking even when I was nursing my first daughter—She was probably getting it too. But we just didn’t know. We hadn’t heard about it. So … I don’t recall if we got results. And if we did—I’m just trying to think now. Why I would’ve continued to breastfeed unless it came back there was, like maybe, a miniscule amount. At that time the thought was nursing was more beneficial, anyway, regardless. And I felt very strongly about nursing. And so maybe even if there had been a miniscule amount or something, I probably would’ve continued nursing anyway. (Resident)	Uncertainty
Physical and emotional impacts of uncertainty: “I think of all those missteps, just from lack of knowledge.”	Yeah, we were concerned because we wondered, the milk we were drinking, was it safe? And meat was a large part of our diet. We wondered about that. We were all fearful. Lot of uncertainty at that time. Many upset people. (Resident) And I think that’s what the whole community even now feels, even as when I go to the class reunion, I think that’s a big conception that it wasn’t handled right and the community’s pain, suffered from it for years now. (Chemical worker family) Yeah, sure. The human side of PBB, that’s what you want to know. There was no. 3 parts per million, like there was with the animals. You either had a disease or you didn’t have a disease. Doctor Selikoff came to Grand Rapids, 1976—god, fall of 1976, about this time of the year, came here and he started, he had a lot of people, [partner name] signed them up and they were hurting and nobody listened to them … its emotional. (Attorney)	Uncertainty

^a^ Cross-Cutting Themes: *Uncertainty,* associated with lack of information or contested interpretations.

### 3.4. Theme Four—Inaction and Loss of Trust in Institutions: “It Was Disconcerting to See How It Was Dealt with.”

This theme explores the implications of governmental inaction for participants’ trust in political and regulatory institutions. As noted earlier, *(in)action* is also a cross cutting theme that appeared throughout multiple themes described in this analysis. Here, we examine explicitly the implications of inaction on the part of state institutions who were perceived to have responsibility for acting to protect the lives and health of Michigan residents and the social and political impacts of that inaction for formerly trusted institutions. Additional quotes for each subtheme are presented in Table 5.

#### 3.4.1. State Inaction: “It Was a Funny Thing That the State of Michigan Did Not Take an Aggressive Stand.”

Reactions to the state of Michigan’s response to the contamination ranged from bewilderment to frustration to anger. A state employee who independently sent eggs from his brother’s sick chickens for laboratory testing and found elevated levels reported: *“I called the Michigan Department of Agriculture, and I mentioned to them I had been to [city]. I saw these sick looking chickens, and I had the eggs checked, and they had PBB, and their response was, ‘Well that’s impossible. There’s no PBB up there in that part of the state. We don’t have any record of it, so we’re not going to do anything about it.’”* Another participant described state inaction in detecting PBB in his dairy herd: *It’s a funny thing that the State of Michigan did not take an aggressive stand. I think they should have. Because the only way we got detected for PBB in our herd wasn’t from the Michigan Department of Agriculture, or the State of Michigan, it was an initiative from our dairy plant that we sold our milk to*.” State inaction was also described regarding recommendations made by a scientific advisory panel appointed by then-Governor Milliken [6]. Following a 10 June 1976 meeting to consider the recommendations to lower the quarantine level, *“the Michigan Department of Agriculture ruled against the panel recommendations. So, people continued to eat these diseased animals, at the levels of 0.3 parts per million for the meat.”* Here, the MDA’s decision not to accept a recommendation from a specifically appointed panel of experts contributed to a loss of trust in state agencies with responsibility for protecting the health of state residents.

#### 3.4.2. Loss of Trust in Institutions: “That Was When I Really Started to Lose Faith in Our Political System.”

Farmers and non-farmers alike described feeling unprotected and unsupported by delays in decision making by the state of Michigan, including, but not limited to the example in the preceding section. One participant described their mother’s attempts to urge action: *“My mother was pretty upset over it; the whole time that it was going on, … She wrote letters to Senator Kennedy … to the governor, several of the main people at the Michigan Department of Health, and several [members of the state] legislature, urging them to come forth and do some health studies and help the farm families,”* and went on to describe their disappointment in the state’s inaction: *“… I cannot fathom why our legislature, our Michigan Department of Health, did not initiate anything on that part. It—it saddens me to no end that it was a very tragic event, but to lessen the burden it would be to help correct the wrongs that happened.”*.

Participants also described disappointment in the state’s failure to hold Velsicol, the plant responsible for the contamination, accountable: *“… it was very disconcerting to see how [the role of Velsicol in the contamination] was dealt with. Because you saw where the chemical plant, basically, was released from any responsibility … And there just seemed to be a scramble for, in a way, a cover-up, not making it sound so bad. Basically, not to alarm people—but we should have been alarmed.”* Delays in state action, and in sharing information, led some participants to conclude that the absence of information was intentional: *“… actually, the Farm Bureau didn’t want us to know what happened. Michigan Chemical [owned by Velsicol] didn’t [either].”*.

Together, delays in state action to protect the population combined with the perception that information was withheld and that responsible parties were not held accountable, began to undermine trust. As one participant noted *“I really started to lose faith in our political system.”* Actions taken by community residents in an effort to promote environmental health protections in the face of inaction are examined in the final theme.

**Table 5 ijerph-19-16686-t005:** Central theme four—Inaction and loss of trust in institutions: “It was disconcerting to see how it was dealt with.”.

Subthemes	Additional Illustrative Quote	Relevant Cross Cutting Theme(s) ^a^
State inaction: “It was a funny thing that the state of Michigan did not take an aggressive stand.”	But nobody mentioned anything about the potential human health effects, until I started talking about my studies … And it seemed like there was really no progress. Several months had passed since the problem was discovered, since I had found out that this stuff has bad effects on pregnancy and—ah—animals … Nobody was listening to what I was finding. The people were still continually eating contaminated meat, milk, and dairy products, and eggs and chickens, and everything. (Doctor) … in 1974 I had raised concerns about women breastfeeding their babies, by June of 1976, the Michigan Department of Public Health announced that—Oh they found PBB in human breast milk! So two years—almost about two years later. (Doctor) But I kind of thought maybe someday they would come back where they want to check blood again, or check stool samples—see if you’re losing it and see what your numbers are. But they never have … To see if anybody like, you would think they’d want to check like, like me when I was in my low teens, growing up. And my levels were at this and let’s check it today and see you know, on the records, and see where it’s at. But they don’t do it, I’m kind of shocked that, you know, that they haven’t done that. You know just to check to see if it is getting higher in your system over the years. But nothing has happened over it, so. But we’re not dying so, I guess, we’re doing all right. (Farm family)	(In)Action
Loss of trust in institutions: “That was when I really started to lose faith in our political system.”	And I wanted to participate in the oral project to state my disappointment with the government of the state of Michigan, which employed me by the way (laughs) for twelve years … where I worked until retirement—I believe Michigan’s response to the PBB conta—poisoning and contamination has been a failure of government. (Resident) And I have never, in my entire life, seen such a dedicated, hard-working—in spite of city obstacles, county obstacles, state obstacles, federal obstacles—that grew from Saint Louis [Michigan], the Pine River Superfund Citizen Task Force. Uhm, they are put on a pedestal in my opinion because they have continued to battle more, often than not, there has been some successes along the forty-year long road, but there’s also been a lot stepping backwards or denial or cover-up. And sometimes all of the agencies—whether they are state or federal—the business itself, the chemical plant, is all part of a, I would call it, greed and a way to, uhm, almost falsify what took place, or deny what took place. (Resident)	(In)Action

^a^ Cross-Cutting Themes: *(In)Action* defined as actions taken (or not taken) in the face of the environmental contamination.

### 3.5. Theme Five—Community Leadership, Knowledge and Skills in Action: “The Community Has to Speak Up about Things.”

As has already been apparent in the preceding themes, community leadership took multiple forms and played a central role in moving action on the part of the state and others. Individual leaders, or champions, played a key role, as did the voice, experience and knowledge of affected community members. Here, we also examine how the community has sustained their efforts some forty years after the contamination. This theme explores the cross-cutting themes of *community skills and resources* and *(in)action*, showing how the community was able to take and sustain action by mobilizing leadership skills and community support to counter the inaction they faced. See full quotes for each subtheme, Table 6.

#### 3.5.1. Leadership, Persistence and Dedication: “It Was One Person … [That] Was Relentless in Finding Out What Was Going On.” 

Participants in this study described several community members who took leadership roles in building information, including Rick Halbert and local veterinarian, Dr. Alpha “Doc” Clark, referenced in earlier sections. One participant described how Rick Halbert’s leadership to build information paved the way for state action, *“Again, it was one person in the state of Michigan—the Halbert family—that went after it, was relentless in finding out what was going on with his cattle being sick. The testing that he went through, the turn-downs that he got, the State of Michigan saying, ‘We don’t have the money’ … And, he kept relentless—he put—spent money, he spent—he spent political capital, he spent his friendship—relationship, networking—to get somebody to test it that actually found that, ‘Hey, you’ve got a heavy chemical in your feed here.’ And then that opened the door to where, then, the state came in and tested it and found out.”*.

#### 3.5.2. Community Support for Action: “Get Up There and Tell Them What’s Happening.”

Beyond these individual champions, other community residents caught in the unfolding contamination began to recognize the delays in state action and encouraged each other to share their experiences and first-hand observations to inform decision making: *“That’s when we were meeting with policy makers and I remember they were having one PBB meeting and … that’s about the time that [redacted] lost her first baby and then was having problems … [They] kept pushing me and pushing me, ‘Get up there and tell what’s happening. Tell them that you’re afraid that your daughter’s [breast milk] is killing your grandchild because you know it’ And I did go up and testify. And I think that’s when they discovered that yes mothers’ [breast milk] was going into these infants.”*.

Participants described their growing mistrust in decision makers, and commitment to build their own evidence base, give voice to their experiences, and hold decision makers accountable for actions taken (or not taken): *“Well we’ve learned that—from way back, from the first cleanup that you can’t trust the agencies to do it right. We’re the voice for the community … we have to ask the questions, we have to see the documents. We have to know for ourselves that something’s being done right.”*.

This emergent leadership was founded in a strong sense of commitment to others in the community, and the conviction that if action were to be taken to clean up the contamination or reduce adverse health impacts, it would occur only through concerted community organizing and action.

#### 3.5.3. Sustaining Action for Change: “They Are Expending Huge Amounts of Time to Volunteer and Drive This Issue and Keep It Alive.”

Sustained community efforts to address the PBB contamination and its aftermath some forty years after the contamination were motivated by a sense of connection or relationship: *“You hate having to hear all those stories of people’s children and miscarriages, and all the things that people have suffered with all these years from being exposed, but at the same time we’re getting it out in the open. Maybe it won’t help most of the people, but maybe it—down the road it will help some. That’s our hope.”* Within this context, participants described efforts to establish scientific and personal knowledge about the health impacts of PBB as an act of both individual and collective action: *“Whatever I can learn, I can spread out and tell other people about it too and make them aware, make them get involved in it too.”*.

These community-led efforts to gather information and build knowledge were used in many instances to organize for change. Initially focused on seeking answers and justice for farmers and impacted families, organizing later also turned toward efforts to clean up contaminated soil in the town surrounding the Velsicol plant and other impacted areas.

Community members developed new structures to support sustained action to address the sources and the implications of the contamination for the health of residents, including, for example, the Pine River Superfund Citizen Taskforce, PBB Action Committee, PBB Citizens Advisory Board and PBB Research Leadership Team. In each of these groups, the presence of experienced, skilled leaders willing to continue to engage in efforts to address health impacts and clean up the contamination were central. One participant shared, referring to the members of the Pine River Citizen Task Force focused on the clean-up of the St. Louis Superfund site: *“They’re not lawyers, they’re not business people, they’re not CEOs. They’re running their life, which is already very busy, but they’re expending huge amounts of time to volunteer and drive this issue and keep it alive. They’ve kept it alive through thick and thin.”*.

Sustained information seeking also played a role in efforts to clean up the contamination. As described by one member of the Pine River Superfund Citizen Task Force (the EPA designated Community Advisory Group (CAG) for the Pine River Superfund Site), their information seeking skills and resources were essential to convincing federal and state actors to take action, decades after the initial awareness of the contamination: *“We had to convince them, we used our money to hire Public Sector Consultant … Their report came back and said, “The plant site is very definitely leaking” so, with that report in hand, we could convince MDEQ and EPA to begin a cleanup.”*.

The persistence of community leaders through the nearly 50-year period since the PBB contamination occurred has been essential to maintaining the attention of state and federal decision makers, and to driving action to remediate the environmental and human impacts. Even so, clean-up of those contaminated sites remains underway, and many Michigan residents continue to experience intergenerational health and economic impacts of those exposures. In the following section, we examine the implications of the findings presented here for understanding the capacity of communities to respond to environmental contaminants in the face of regulatory frameworks that provide, at best, limited support for the protection of population health.

**Table 6 ijerph-19-16686-t006:** Central theme five—Community leadership, knowledge and skills in action: “The community has to speak up about things.”.

Subthemes	Additional Illustrative Quotes	Relevant Cross Cutting Theme(s) ^a^
Leadership, persistence and dedication: “it was one person … [that] was relentless in finding out what was going on.”	Doc Clark, you know Doc Clark? Another hero. Had my pickup truck and I was supposed to pick him up some time out in the middle of nowhere and so on. So I got down this dirt road to pick him up and take him, just to some meeting. I didn’t know him that well and he said, well, he said, he just did another PBB sample—he actually cut up behind the ear, behind the ear was a good place to get fat. He said, “Just did another cow.” I said, “geez, I didn’t think anybody was dead.” And he said, “Neither did I, this cow just dropped over today.” And so he took the fat sample right there. He and—well, Rick Halbert is the reason you and I are here today. (Attorney) … all of these fat samples Doc has taken, and probably other vets that we were sending to MSU [Michigan State University] come back clean. And the cattle were still showing the very same symptoms that they always had. So Doc—he’s pretty clever—we started taking two fat samples; he’d send one to MSU, and he’d send the other to a place in Wisconsin to a place called WARF [Wisconsin Alumni Research Foundation], I don’t—maybe you’ve heard of it, or know of it? The one from Doc would come back contaminated, and the one from MSU would come back clean. (Farmer) … it just takes one person with the diligence and the—and the fortitude to say, “I am right, and you are wrong, and I am going to fight this until the end.” And lay everything out on the line; these people laid everything out on the line to come up with this. (Farmer) Don Albosta who was our representative and the senator from Quincy, came up hunting—and this is during the time when we were going through this because we took hunters in … And, that he realized too, we were trying hard, and we loved our cattle that he really stepped up forward, too. He went to bat, so to say for the farmers, you know. (Farm family) Hopefully it will stay somewhere. I mean, maybe y’all can take it from me, I don’t know. (laughs) Because I have on Facebook I have a support group and a lot of people have shared stories there … Yeah, that’s why this isn’t … It’s not a company. It’s not an organization. I organized it the way I organized it for a reason. (Resident)	Community skills and resources (In)Action
Community support for action: “Get up there and tell them what’s happening.”	But we are all in this together. And right now it is very important to me to be part of something that is truly a important mission. And I really passionately view that … You know, just to hear their story and to see the emotions in their face when they’re telling you this. And how could you not listen to that and understand something horrible has happened? We can’t ignore this any longer. (Resident) But we went to Lansing. To try and get them to realize that it’s not just, it’s not just the farmers. And I went to that PBB meeting where I testified for my granddaughter and my daughter nursing her. And then we went to the Tacoma trial as much as we could. Taking turns to drive so we wouldn’t have that much money in gas and eating peanut butter sandwiches. And we sold gravel and trees trying from our farm to hold out till after the Tacoma trial thinking we’d get our turn in court, but I don’t know, there’s 80 some farms I think that never even got into court. (pause) So. I don’t know. (Farm family) So that’s why [the mayor] started coming to the meetings, and he actually apologized to me, he said, “you guys were right and I was wrong.” And I said, “wow, thank you for saying that. It’s very good to hear and I hope you’ll keep coming” and all that. And they did. So that—they turned the page there. And then we all started working together. (Resident) Margaret Mead that said that small groups of people do change the world. I’m paraphrasing, but it’s true. They do. And—and I have this whole thing about one person can make a difference. One person really can both good and bad. But for good is what I try to focus on. And I think I’ve always been drawn to that idea about people working for the good of—of the community. (Resident)	Community skills and resources (In)Action
Sustaining action for change: “They are expending huge amounts of time to volunteer and drive this issue and keep it alive.”	I guess I need to communicate also that there’s been a monumental amount of change and clean-up that has already taken place. And it isn’t from anything that I have done directly. It’s from the work that the local people have done on that CAG [Pine River Superfund Citizen Task Force]. They have been relentless. (Resident) And it’s not that the community hasn’t tried. We have the CAG [Pine River Superfund Citizen Task Force], which is the community action group that was set up by the EPA. It’s the longest running one ever. And technically I would consider it the most successful one ever because it’s the only one that’s ever even gotten to go to trial, to go to court and it actually won money. (Resident) Some of the people that I met from St. Louis and Alma, to know within their own personal families what tragedies took place. And they still come to the table and they (pounds on table) still are there. And they’re the ones who are the heroes. They’re the ones who are the heroes. They never gave up. (Resident) I was asked by a sociologist years ago, probably nine years ago now, “how does your group just keep going? What’s different about you?” I said, “I have no idea, I thought every group was like this” you know? She said, “no, most groups that are gathered around a cause of any kind last no longer than seven years, it’s like a rule in sociology. A community group, volunteers, gathered around a cause don’t last any longer than seven years. We were already at 10 years at that point. And, so she started doing interviews with us to try to figure out (laughs) what made us different. And I liked her one theory, she said several of the people involved in our group not only have lived here their whole lives, but their parents came—mine were the earliest, my ancestors were the earliest, but a lot of them came in the late 1800s, early 1900s, and settled here. So, she thinks that’s part of it, that we just have our roots here, but also that we have the pioneer spirit, isn’t that interesting? (Resident)	Community skills and resources (In)Action

^a^ Cross-Cutting Themes: The mobilization of *community skills and resources*, defined as human or organizational capital to address the environmental health issue. *(In)Action,* defined as actions taken (or not taken) in the face of the environmental contamination.

## 4. Discussion

Michigan’s PBB contamination created profound threats to livelihoods, ways of life, and ultimately health and life itself. Michigan residents caught in the path of this environmental catastrophe sought to protect themselves and their communities under conditions of ambiguity, uncertainty, and in some cases, deliberate efforts to contest and obscure local knowledge and experience. Understanding the processes through which individuals sought information, critically analyzed complex and sometimes contradictory messages, and acted to protect health and well-being in the face of the large-scale PBB contamination and in the ensuing decades, offers much to the field of environmental health.

The themes presented in the preceding section begin to disentangle this complex experience. In this discussion section, we consider contributions of the above analysis to our understanding of environmental health more broadly. In doing so, we draw on three conceptual frameworks relevant to understanding how communities affected by environmental threats engage to protect their health: environmental health literacy [15,22,23]; community capacity for environmental health promotion [16]; and “contested illnesses” [17]. Together with the themes that emerged from our analysis, we use these frameworks to build an integrated conceptual model that considers environmental health literacy embedded within broader social processes that shape the capacity of communities to protect themselves against emergent environmental health threats.

Community residents described their initial awareness that *something was wrong*, based on their observations of farm animals. Their own efforts to understand more clearly the source of the problem and the geographic scope of affected farms met with a variety of responses, including lack of clear information, misinformation, dismissal of concerns, and attribution to their own animal husbandry techniques. In the absence of clear environmental health science, and in the face of contested explanations, many turned to building their own evidence. Members of the affected community possessed important knowledge and skills to recognize and understand that something was wrong; that knowledge was too often discounted or contested. Challenges to the knowledge and experience of community members included government and industry efforts to discredit their observations of their animals and their own health. It included explicit efforts to question the legitimacy of their experientially based knowledge and to blame the farmers for the environmental catastrophe they were experiencing. Those challenges undermined the political will to fully protect the health and economic interests of community members.

Recognizing the importance of valuing and building from community knowledge, in a critique of the environmental health literacy literature, Cole noted that the “centrality of scientific content to environmental literacy is problematic” [23] (p. 35). This critique is echoed in Finn and O’Fallon [15] who note the linked assumptions within many EHL frameworks: (1) that environmental health science is present and available; (2) that environmental health science is central to the ability to take action, and (3) that the ability to act is determined by the individual’s level of EHL. In this PBB case study, these assumptions were not met. Specifically, formal scientific information was not initially available in the early stages of unfolding awareness and action. The slow pace at which scientific knowledge emerged, the contested nature of that knowledge, and prevailing frameworks demanding a high degree of scientific certainty before action is taken, created important challenges for individual and state actors faced with the need to take immediate action. Yet few EHL models offer analytic tools that challenge the linear assumption that formal scientific knowledge emerges first and then is transferred to and acted upon by other users, or for addressing the sociopolitical context within which environmental health contaminations, and community efforts to protect their health, emerge. One such tool is the precautionary principle [24], which urges decision makers to take preventive action in the face of uncertainty. The precautionary principle also encourages decision makers to shift the burden of proof from communities experiencing harm from exposure, to instead requiring industries that produce a chemical or product to first prove that it is safe. In Michigan’s PBB contamination, both individual and state actions were repeatedly delayed in the face of uncertain and contested information regarding the harms caused by the exposure, rather than requiring Velsicol to prove that PBB did not cause harm.

These limitations of traditional EHL models are particularly salient given the cross-cutting themes that wove through the analysis presented here. Those themes included the pervasive impacts of *uncertainty,* connected both to a dearth of information and contested interpretations of the information that was available; the implications of that uncertainty for delays or failure to take action *((in)action)*; and the mobilization of *community skills and resources* in an effort to build knowledge to inform action to protect health. We found the work of Phil Brown [17] on contested illnesses and the environmental health movement, and that of Nicholas Freudenberg [16] on community capacity to promote environmental health, to be particularly relevant in understanding these cross-cutting themes, and discuss each below.

*Contested illness, uncertainty, and (in)action*. The concept of contested knowledge that emerged from the oral histories of residents affected by the PBB crisis is relevant to the literature on contested knowledge and contested illness. The social theory of contested knowledge recognizes the relationship between knowledge and power [25], which we saw in the experiences of these residents, who were left to make difficult moral decisions with a lack of information amidst uncertainty, inaction, and power imbalances. Their experience especially resonates with the theory of contested illnesses, which emerged in relation to environmental health movements to characterize illnesses whose connections to environmental exposures are challenged, often in the face of uncertain, ambiguous or incomplete scientific data and in the context of high stakes political struggles over the source of exposures and responsibility of corporate entities [17]. In the PBB case, the absence of clear, certain, and timely scientific information combined with a politicized decision-making context left individuals to make decisions on their own, and contributed to delays in individual and collective action to protect health. Examples include accusations that farmers were to blame for the loss of their livestock due to poor animal husbandry practices, which diverted attention from the widespread nature of the contamination as well as the industry responsible for it. Furthermore, the MDA permitted farmers to sell farm products if their animals fell below the actionable level of PBB, leaving many low-level farmers with the difficult decision of knowingly selling contaminated food goods or facing financial ruin by destroying their herds. Similarly, the MDPH did not issue a formal warning about breastfeeding, but instead issued a statement that it was up to the mothers to decide, and that no toxic effect had been demonstrated in human milk [2].

Similar to the “contested illnesses” described by Brown [17] (e.g., Gulf War related illnesses), symptoms experienced by the animals and humans exposed to PBB were diffuse and their links to environmental exposures uncertain, with some appearing many years following the exposure. The difficulty and long timeline required to establish causality, lack of funding, and lack of consistency across studies contributed to the long timeline required to address scientific uncertainties. This, combined with power relations and the economic interests of the corporation involved with the contamination, created a scenario in which links between the environmental contamination and its health effects could readily be contested and challenged. The resulting uncertainties contributed to delays in health protective actions and left the onus of decision-making on the impacted individuals. This interplay of contested knowledge, illness, uncertainty, and inaction again echoes support for the precautionary principle in shifting the onus of decision-making to those in power, despite uncertainties.

*Community capacity to promote environmental health*. In 2004, Nicholas Freudenberg and colleagues [16] adapted frameworks examining community capacity as a general public health construct [26] to specifically examine factors that influence the capacity of communities to protect themselves in the face of environmental threats. These frameworks offer a model for centering community action to protect and promote health—including environmental health—and are an important advance over frameworks that assume linear processes of literacy leading to action. Specifically, they recognize multiple interconnected components of communities as social systems with synergistic characteristics that shape their response to environmental and other threats to health and well-being. Multiple scholars have since built on that framework to consider the ability of communities to mobilize and influence public decisions to promote environmental health [27,28,29,30].

Components of community capacity include, for example, community skills, participation, leadership, a sense of community (defined as a sense of shared history, place, or identity), and community power [16,26]. In most writing on community capacity, these characteristics are conceptualized as both latent potential and actualized manifestations of community capacity. These models recognize that capacity emerges within historical contexts, is cyclical and iterative, and is modified over time [16,27]. In other words, latent dimensions of community capacity exist within communities and can be called upon or mobilized to protect health in the face of environmental threats. Those dimensions can be enhanced or strengthened in the process of working together to promote environmental health. They can also be eroded or undermined.

The above frameworks, together with the analysis presented above, inform our conceptual model shown in Figure 1. This model illustrates the mobilization and strengthening of latent community capacity over time, creating environmental health knowledge to inform action. Reflecting the themes emerging from the PBB oral histories, this conceptual model centers latent or existing community capacities and attempts to capture the cyclical and non-linear process through which those capacities were mobilized to promote environmental health.

The initial environmental exposure, the PBB contamination, is conceptualized as the catalyst that sets in motion the translation of latent or existing community capacities. Those existing capacities, represented in the inner, light blue circle, are mobilized into actualized capacities, represented in bold text and in the circles around the perimeter of the model. Central among these are existing skills, which Freudenberg [16] defined broadly as “… relevant organizational, scientific, political, and information seeking skills among a range of participants” [16] (p. 474). As shown in Circle 1 (Figure 1), within the farming communities first impacted by PBB, residents’ existing skills in animal husbandry were central to the early recognition that “there is something wrong.” Yet, these skills were challenged, as explored in the theme “They were telling people we didn’t know what we were talking about: contesting community knowledge.” The resulting uncertainty contributed to frustration on the part of those directly affected, and to inaction on the part of decision makers.

Uncertainty and inaction in the face of threats to their livelihoods and health served as a catalyst for community members to mobilize existing community skills, resources, and leadership (Figure 1, Circle 2). In early years, community members mobilized skills in chemistry and veterinary medicine to conduct their own research, seeking to build a body of evidence to bolster their initial observations and inform understanding of the source of the contamination, its links to animal disease and deaths, and its geographic scope. As these efforts progressed, the skills mobilized were not limited to scientific and medical skills, but extended to encompass leadership in political and organizational efforts. Leadership, another dimension of community capacity, was both formal and informal, from farm families and factory workers affected by the contamination to medical, scientific and legal professionals embedded within the community. As noted in the community capacity literature cited previously, this process was not linear, but unfolded in a cyclical and iterative manner.

As community members mobilized their skills, they were also faced with the limitations of the information available to them, and their own capacity, noting continued uncertainty, inaction, and challenges to their knowledge. In response, they mobilized another dimension of capacity, social and organizational networks and resources, to seek outside expertise to complement community-driven research, reflected in Figure 1, Circle 3. They mobilized social networks including connections with family members, friends, and professional networks with medical, epidemiological, toxicological and legal expertise, as well as political networks. Focused substantially on building the body of knowledge and understanding about the contamination, its sources, health impacts, and geographic distribution, the community skills that were mobilized in this process extended beyond knowledge building to include leadership, participation, and political networking skills.

This model reflects the “community leadership, participation, and sense of community” (Figure 1, Circle 4) that was actualized during the PBB contamination. As community members built knowledge and understanding of the PBB exposure, its implications, and its scope, it became increasingly clear that information alone would not be sufficient to create change to promote community health. Rather, continued leadership and participation were central to sustain research to understand the health impacts of environmental exposure, maintain the attention of elected and other decision makers, and demand action to address this large-scale chemical contamination. The mobilization of community leadership and participation was apparent as participants connected with others affected by the PBB crisis, and encouraged each other to participate in political and decision-making processes as in the theme “the community has to speak up about things.” Through such efforts, participants also built on and strengthened a sense of community or shared fate, another key dimension of community capacity [16].

Circle 5 in Figure 1 reflects sustained leadership, skills, resources, and participation that manifest as community power more than 40 years after the initial PBB contamination. Captured in the theme “they are expending huge amounts of energy to keep it alive”, the sustained participation and leadership that is evident throughout these narratives began with a commitment to building knowledge, and extends to persistent engagement in the use of that knowledge to redress community harms and strengthen future protections to promote environmental health.

The cyclical process captured in the conceptual model, together reflects the emergence of community power, the final dimension of community capacity that we highlight in this analysis. Defined as the community’s ability to act, to make or resist change that affects their environment [16,26], community power is central to the PBB story. Community power involves access to information on which to base decisions and action [16], particularly relevant given the initial limited power of the PBB community to act due in part to challenges in obtaining and interpreting information with which to inform action. Indeed, contested knowledge, uncertainty, and inaction undermine the community’s capacity to protect their health and well-being in the face of this environmental threat. They served as catalysts for the mobilization of community skills to build knowledge and to apply that knowledge to action, and highlight the limitations of reliance on scientific knowledge alone to inform action. Indeed, dominant frameworks which require a high level of scientific certainty to inform policy making too often impede action in circumstances in which sparse information, political contestation of the information available, and the uncertainty and ambiguity of the information that is available delay decisions and action.

Lessons learned, for those working with communities to promote and strengthen capacity to protect health in the face of environmental threats, include the importance of moving beyond traditional conceptualizations of EHL that center scientific environmental health knowledge toward a more nuanced understanding of the social and political factors that shape access to information and its application to health-promoting actions.

## 5. Conclusions

Paul Slovic, writing over two decades ago, states that “Whoever controls the definition of risk controls the rational solution to the problem at hand … Defining risk is thus an exercise in power” [31] (p. 689). This quote resonates with the experience of community members affected by the PBB contamination. Contested explanations of the source of the exposure, whether individuals or corporations were responsible for the health impacts that emerged, and even the extent to which those health risks were “real” were all manifestations of efforts to control the definition of risk, and relatedly, responsibility for action to protect health. The community’s power to protect their health from this environmental threat was hampered by an absence of scientific knowledge, but more importantly by efforts to devalue and undermine the knowledge drawn from their own observations and insights, and by frameworks for action that require proof of harm before preventative action is taken. This analysis emphasizes the importance of environmental health practice that supports and sustains community capacity for action to protect health, and a reframing of policy efforts to include a precautionary approach that emphasizes protections against likely harms rather than delays in action until harm has been scientifically “proven.” Efforts to improve outcomes must consider more than an individual or community’s level of EHL, but also the sociocultural factors of existing community capacity and power dynamics, particularly around information. Continued efforts to promote environmental health knowledge must clearly examine and contest the power dynamics that center and privilege professionally produced scientific knowledge.

This analysis of narratives from Michigan residents who encountered, and survived, the large-scale PBB contamination in the 1970s captures the lived experience of those residents, and informs efforts to understand the role of environmental health science in shaping actions to promote health. It highlights the limitations of linear, decontextualized frameworks that emphasize academic research without consideration of the context within which it is placed. Results from this analysis of the Michigan PBB contamination encompass the social context within which an understanding of exposure and associated health risk emerged. Importantly, they highlight the critical role of community as well as academic science in this process. Our findings suggest the importance of integrated frameworks for examining and promoting the critical role of community skills, leadership, participation, sense of community, and community power in promoting environmental health. Failure to consider those dynamics offers at best a partial analysis of the forces that shape not only environmental health literacy, but of the agency of community members in creating knowledge and assuring its translation to action to promote health. There is a critical need for additional research that examines the interplay of professionally produced science with community capacity and power imbalances, and for environmental health promotion that emphasizes precautionary approaches and builds community capacity toward the goal of improving environmental health protections.

## Figures and Tables

**Figure 1 ijerph-19-16686-f001:**
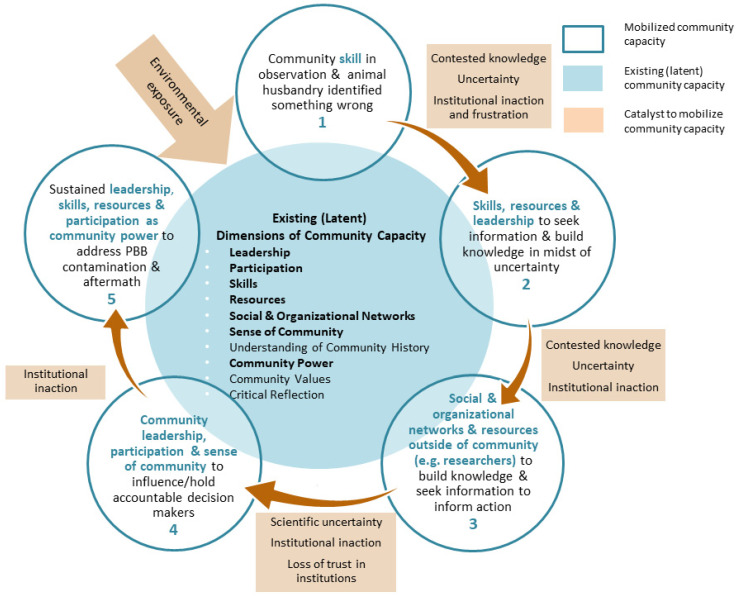
PBB Oral History Conceptual Model. Informed by Freudenberg [16], this model centers community capacity that is mobilized by an environmental exposure and other catalysts.

**Table 1 ijerph-19-16686-t001:** PBB Oral History Qualitative Themes.

Central Themes and Subthemes	Relevant Cross-Cutting Themes
**1.** **Contesting Community Knowledge**	
Personal observation of animals	Community skills and resources
Contested nature of knowledge about the contamination.	Uncertainty
Inaction and frustration	(In)Action
**2.** **Seeking information and building knowledge and understanding in the midst of uncertainty**	
Leveraging community expertise to build knowledge	Community skills and resources
Seeking information outside the community	Community skills and resources; (In)Action; Uncertainty
**3.** **Human impacts of uncertainty**	
Uncertainty and ambiguity regarding health guidance	Uncertainty
Physical and emotional impacts of uncertainty	Uncertainty
**4.** **Inaction and loss of trust in institutions**	
State inaction	(In)Action
Loss of trust in institutions	(In)Action
**5.** **Community leadership, knowledge and skills in action**	
Leadership, persistence and dedication	Community skills and resources; (In)Action
Community support for action	Community skills and resources; (In)Action
Sustaining action for change	Community skills and resources; (In)Action

## Data Availability

The oral histories are not yet publicly available, as they are being prepared for donation to the repository, the Museum of Cultural and Natural History at Central Michigan University. Interview materials may be accessed via written request to the Michigan PBB Oral History Project Director, Brittany Fremion at fremi1b@cmich.edu.

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
