# Peer review of "“They Kept Going for Answers”: Knowledge, Capacity, and Environmental Health Literacy in Michigan’s PBB Contamination"

_ijerph, 2022, doi:10.3390/ijerph192416686_

Round 1
Reviewer 1 Report
This is a very interesting manuscript which reports on a very important environmental incident of contamination that brought so many negative environmental, social and health impacts to a significant number of people and their livelihoods.
The paper is based on the Michigan PBB Oral History Project which has documented community residents’ descriptions of a large-scale chemical contamination – the PBB contamination – that occurred in Michigan in 1973.
While the paper is information or text-intensive due to the methodological approach chosen and the analysis is good, there are some minor weaknesses that must be addressed in order to improve the overall presentation and the scientific approach. My specific comments are summarised as follows:
#Title of the manuscript: Comment: The title appears relatively long (i.e. 21 words in total) and can you please rephrase and remove abbreviations in the title. Furthermore, if this is difficult to achieve, I am suggesting that you delete the first part currently written as: “They kept going for answers”.
#Abstract: Comment: Your abstract does not suggest or give the reader any idea about what results were revealed by the study. This can be done by just one sentence to give a broader overview of what came out of this study.
#Introduction: Comment: In your opening introduction, you preface the introduction with a very captivating prelude or preface and this is in the form of a 'quotation'. However, considering the originality issues of your manuscript, my question is that why is this quotation not referenced? If this is your quotation then there is no problem, but if this comes from another source, you need to give due credit to the original source by writing it down as well so that we all know where does the info come from?
#Referencing error: Comment: I read the following statement that is written as follows:
The remaining transcripts were coded by one of the team members (rather than two), following protocols suggested by Fleiss, Levin, Paik (2003) [21].
While the referencing appears accurate, I don't think that the '(2003)' part has to be written as well and therefore this information must be deleted.
Furthermore, the same mistake appears in Line 612 where the year '2017' must be replaced with a number that represents this literature in the List of consulted references provided at the end of this manuscript. The affected sentence read as follows:
This critique is echoed in Finn and O’Fallon (2017) who note the linked assumptions within many EHL frameworks: 1) that environmental health science is present and available; 2) that environmental health science is central to the ability to take action, and 3) that the ability to act is determined by the individual’s level of EHL.
#Presentation of results: Comment: In dividing your sections or subsections, the manuscript is using capital letters and sometimes certain phrases to capture the various themes are being underlined while others are not being underlined. It is not clear why are we seeing this pattern? My comment is that capital letters should not be used throughout as titles of subsections and the underlining of certain texts must be removed as well. However, if you cannot remove the underlining pattern, then the reader needs to know why?
#Conclusion section: Comment: The conclusion part is very brief but very interesting to read. However, it can be expanded a little bit but with not more than 5 extra lines to give an overall context of what this study has accomplished and brought to the fore. Lastly, it will be very helpful that this part of the manuscript also offers to the readers a few recommendations to suggest areas that still needs to be researched in the future.
I hope my comments are understandable and helpful towards improving the readability of this insightful manuscript.
Good luck.
Author Response
Please see attachment. Thank you for your review!

Reviewer 2 Report
“They kept going for answers”: Exploring contested 2 knowledge, community capacity, and environmental health lit-3 eracy in the Michigan PBB Oral History Project 4
Erin Lebow-Skelley1*, Brittany B. Fremion2, Martha Quinn3, Melissa Makled3, Norman B. Keon4, Jane Jelenek5, 5 Jane-Ann Crowley5,6, Melanie A. Pearson1 and Amy J. Schulz3
This article documents an early episode in environmental contamination and one that probably has contributed to citizen distrust of government to protect and science to explain. I hope the authors go on to produce a case-study of this event, which would include the timeline/state of knowledge of, ability to test for, and toxicity of PBB.
How were categories named? Were names in the literature assigned after the concepts were identified (ala Freudenberg)? I understand it is the prerogative of the authors to establish continuity with existing theoretical exercises, however it would be great to have a peek into their thinking and thus the connection and embellishment of the “model” pictured in Figure 1. Could this be described in the methods?
Knowledge conflicts are not well defined. Does it mean disagreements between “scientists” and “citizens”? There were scientists. As those at MSU who did the testing? Outside veterinarians? “independent lab”… There were also citizen scientists? Rick Halbert-chemist, Doc Clark, and others. Could this be included in the methods?
Page 15 “PBB conta –poisoning” Is this a misspelling?
Page 16, “more, often than not, there” seems garbled, is it correct?
Why are some individuals named and others not? Was it a matter of permission at the time of interview consent? Mention in the methods.
Page 16, is gaining an example that showed perhaps some truth to the concerns that people had as in “…And then that opened the door to where, then, the 513 state came in and tested it and found out.” also a theme? The answer may interest the readers.
Page 19, CAG is an abbreviation, what does it stand for? I see PBB Citizens Advisory Board but no community or citizens advisory group, which comes to mind when I see CAG.
Page 19 “…and the still are there.” Is “the” “they” here. If it’s from the transcript annotate with “(sic)”. The journal editors should make a decision as to how to handle lack of clarity when it comes to verbatim interviews.
Conclusions, page 20, third paragraph, “delegitimize experience” is offered instead of delegitimize knowledge or knowledge conflicts…how are these different/related? Is the connection not made from agriculture to environmental health (literacy) because these worlds did not mingle? What does this mean for the citizen in this situation? Clarity is needed here. Define “contested knowledge” since it seems that the knowledge contested is citizen knowledge.
Page 21, line 663, edit this phrase: “that it it was up to mothers..”
Page 22, line 678, Freudenberg is not the only author. Also the date and number of the citation are incorrect. Should be #16.
Spell out WARF’s full name (in one of the interviews…).
I am puzzling over the idea that from the citizen’s perspective, they had no “science” interactions that came up as a theme though indeed there were several individuals with science (not research) expertise. For instance the MSU failure to id PBB; WARF’s helpfulness; the partnership with a retired chemist and maybe others. It might be worth mentioning the criteria for inclusion in a theme and specifically how this did not warrant inclusion. (Maybe in methods and/or conclusions)
Author Response

(The authors gave the same response as above.)
